# Olfactory receptor OR2AT4 regulates human hair growth

Jérémy Chéret[1], Marta Bertolini[1,2], Leslie Ponce[1], Janin Lehmann[1], Teresa Tsai[3], Majid Alam[1], Hanns Hatt[3] & Ralf Paus[4,5]

Olfactory receptors are expressed by different cell types throughout the body and regulate physiological cell functions beyond olfaction. In particular, the olfactory receptor OR2AT4 has been shown to stimulate keratinocyte proliferation in the skin. Here, we show that the epithelium of human hair follicles, particularly the outer root sheath, expresses OR2AT4, and that specific stimulation of OR2AT4 by a synthetic sandalwood odorant (Sandalore®) prolongs human hair growth ex vivo by decreasing apoptosis and increasing production of the anagen-prolonging growth factor IGF-1. In contrast, co-administration of the specific OR2AT4 antagonist Phenirat® and silencing of OR2AT4 inhibit hair growth. Together, our study identifies that human hair follicles can engage in olfactory receptor-dependent chemosensation and require OR2AT4-mediated signaling to sustain their growth, suggesting that olfactory receptors may serve as a target in hair loss therapy.

[1] Monasterium Laboratory, Skin and Hair Research Solutions GmbH, 48149 Münster, Germany. [2] Department of Dermatology, University of Münster, 48149 Münster, Germany. [3] Department of Cell Physiology, Faculty Biology and Biotechnology, Ruhr-University Bochum, 44801 Bochum, Germany. [4] Centre for Dermatology Research, MAHSC and NIHR Biomedical Research Centre, University of Manchester, Manchester M13 9PT, UK. [5] Department of Dermatology & Cutaneous Surgery, University of Miami Miller School of Medicine, Miami, FL 33136, USA. Correspondence and requests for materials should be addressed to R.P. (email: ralf.paus@manchester.ac.uk)

Olfactory receptors (ORs) are part of an evolutionarily ancient chemosensory signaling system that long pre-dates the development of smell sensation (olfaction)[1–7]. OR expression is not restricted to the nasal epithelium, but it is also present in several other human tissues[8–15]. Non-olfactory roles of ORs have also been described in human cell physiology[16], such as in spermatozoa[10,17] and enterochromaffin cells of the gut[18].

Interestingly, several ORs are also expressed in human epidermis[19,20], including OR2AT4, whose selective activation by the synthetic sandalwood odorant (Sandalore®) promotes human epidermal keratinocyte migration and proliferation in vitro and wound re-epithelialization ex vivo[20]. Sandalore®-induced Ca2+ signaling could be blocked in OR2AT4-transfected Hana3A cells when this was co-applied at equimolar concentrations with the potent competitive OR2AT4 antagonist in presence of Sandalore®, Phenirat®[20]. Given the intimate connections between hair growth and wound healing[21–24], we hypothesized that this OR might also impact on human hair growth. This hypothesis was investigated by immunohistology, qRT-PCR, western blot, microarray, phospho-kinase assay, and gene silencing in healthy, organ-cultured human scalp hair follicles (HFs)[25].

The present study shows that human HFs express a specific OR, namely, OR2AT4. The activation of this OR by its specific agonist, Sandalore®, prolongs anagen maintenance ex vivo by decreasing hair matrix keratinocytes apoptosis and increasing the production of IGF-1 in the outer root sheath (ORS). The anagen-prolonging effect mediated by Sandalore® is OR2AT4 dependent, as confirmed by co-administration of Sandalore® with the OR2AT4 competitive antagonist, Phenirat®, as well as the specific knock-down of OR2AT4 in human HFs. Taken together, we show that human HFs can engage in chemosensation and that the specific activation of OR2AT4 is required to sustain HF growth.

## Results

**Human HFs express OR2AT4.** Immunofluorescence microscopy, qRT-PCR, and western blot analysis revealed that human scalp HFs in the anagen VI stage of the hair cycle[26,27] express OR2AT4 at the transcript and protein level (Figs. 1a, 2a–f). Interestingly, OR2AT4 protein was predominantly expressed by suprabulbar keratinocytes of the proximal ORS (Figs. 1a, 2c), while hair matrix keratinocytes also expressed low-level OR2AT4 protein (Figs. 1a, 2b), both in healthy scalp skin in situ[27] (Fig. 2a–c) and in amputated micro-dissected anagen HFs ex vivo[25,26] (Fig. 1a). Of note, OR2AT4 expression was downregulated during spontaneous, apoptosis-driven HF regression (catagen)[26,27] (Fig. 2d–g). Thus, using the primary antibody employed here[20], intrafollicular OR2AT4 expression is strikingly restricted to defined epithelial HF compartments and is hair cycle dependent.

**OR2AT4 activation by Sandalore® prolongs anagen ex vivo.** When microdissected, organ-cultured human HFs[25] were treated with Sandalore® (500 μM, for details, see Supplementary Note 1 and Fig. 3a–e) for 6 days, this selective OR2AT4 agonist[20] significantly upregulated intrafollicular OR2AT4 protein expression (Fig. 3d), demonstrating receptor functionality and that OR2AT4 expression underlies a positive feedback regulation.

Importantly, Sandalore® treatment retarded spontaneous HF regression (catagen development)[26,27] ex vivo (Fig. 1b) and significantly reduced hair matrix keratinocyte apoptosis, as shown by quantitative (immuno-)histomorphometry for TUNEL+ (Fig. 1c) or cleaved caspase 3+ cells (Supplementary Fig. 1a) in the hair matrix. These effects were partially counteracted by co-administering the competitive OR2AT4 antagonist, Phenirat®[20], with Sandalore® (Fig. 1b, c, Supplementary Fig. 1a). When tested alone, Phenirat® tended to be weakly hair growth inhibitory (Supplementary Fig. 2a, b and Supplementary Note 2 for extended discussion).

Next, we examined two key growth factors that control the anagen-catagen transformation during human HF cycling, i.e., catagen-promoting TGF-β2 and anagen-maintaining IGF-1; these growth factors are prominently produced by those proximal ORS keratinocytes[28–33] that express OR2AT4 maximally. This analysis revealed a significant decrease in TGF-β2 (Supplementary Fig. 3a) and a significant increase of IGF-1 (Fig. 1d) protein expression in the proximal ORS after long-term Sandalore® treatment ex vivo. The co-administration of OR2AT4 antagonist, Phenirat®, significantly reversed the Sandalore®-induced intrafollicular upregulation of IGF-1 (Fig. 1d) but did not affect TGF-β2 expression (Supplementary Fig. 3a).

**Anagen-prolonging effect of Sandalore® is OR2AT4 specific.** Subsequently, we selectively silenced OR2AT4 by siRNA administration to organ-cultured human scalp HFs ex vivo[32,34,35], as documented by significantly reduced intra-follicular OR2AT4 mRNA and protein expression (Fig. 4a, b). Despite the presence of excess ligand (Sandalore®), OR2AT4 knock-down significantly promoted catagen induction compared to HFs treated with scrambled oligos (Fig. 5a), decreased IGF-1 protein expression (Fig. 5b), and enhanced hair matrix keratinocyte apoptosis (Fig. 5d, e). Instead, hair matrix keratinocyte proliferation (Fig. 5c) or TGFβ2 protein expression in the ORS (Supplementary Fig. 4a) remained unaffected. These data show that the Sandalore®-induced hair growth stimulation documented above is indeed OR2AT4 dependent, rather than due to off-target effects of this synthetic odorant and that OR2AT4 signaling is required for anagen maintenance.

**Sandalore®-mediated HF response involves different pathways.** Microarray analysis independently confirmed anti-apoptotic effects of Sandalore® (Fig. 6, Supplementary Fig. 5, and Supplementary Data 1), since transcripts of pro-apoptotic genes were significantly downregulated (e.g., *TP53AIP1*: -10.27×), while anti-apoptotic genes were significantly upregulated (e.g., *FGF-2*: +7.83×) in HFs treated short term with Sandalore® (6 h, Supplementary Fig. 5a, b and Supplementary Data 1). Interestingly, an additional microarray analysis of organ-cultured scalp HFs in which OR2AT4 had been knocked down ex vivo showed that transcription of the *IFI6* (G1P3) gene, whose silencing increases keratinocyte apoptosis[36], was downregulated by administering OR2AT4 siRNA for 6 h, compared to scrambled oligonucleotide-treated HFs (Supplementary Fig. 6a, b and Supplementary Data 1). This corresponds well to our observation that OR2AT4 silencing increases apoptosis of HF matrix keratinocytes (Fig. 5d, e) and further underscores the importance of continued OR2AT4 stimulation by as yet unknown endogenous ligands to suppress apoptosis in the hair matrix of human anagen HFs.

In addition, microarray analysis revealed that Sandalore® promotes signaling along the IGF pathway (see Fig. 6a–c, Supplementary Fig. 5a-b, and Supplementary Data 1), in agreement with the protein expression data (Fig. 1d). Indeed, genes involved in IGF1R signaling cascade as well as in IGF transport (e.g., *PAPPA* [10.8× upregulated] that cleaves IGFBP4 to release IGF[37], or *PCSK-1* [33.6× upregulated] which is involved in insulin synthesis from proinsulin[38]) were strongly upregulated (Fig. 6a–c, Supplementary Fig. 5, and Supplementary Data 1). This promotion of IGF signaling pathway as well as the upregulation of *FGF-7* (2.75× increase), another anagen-promoting growth factor[39] (Fig. 6a–c, Supplementary Fig. 5,

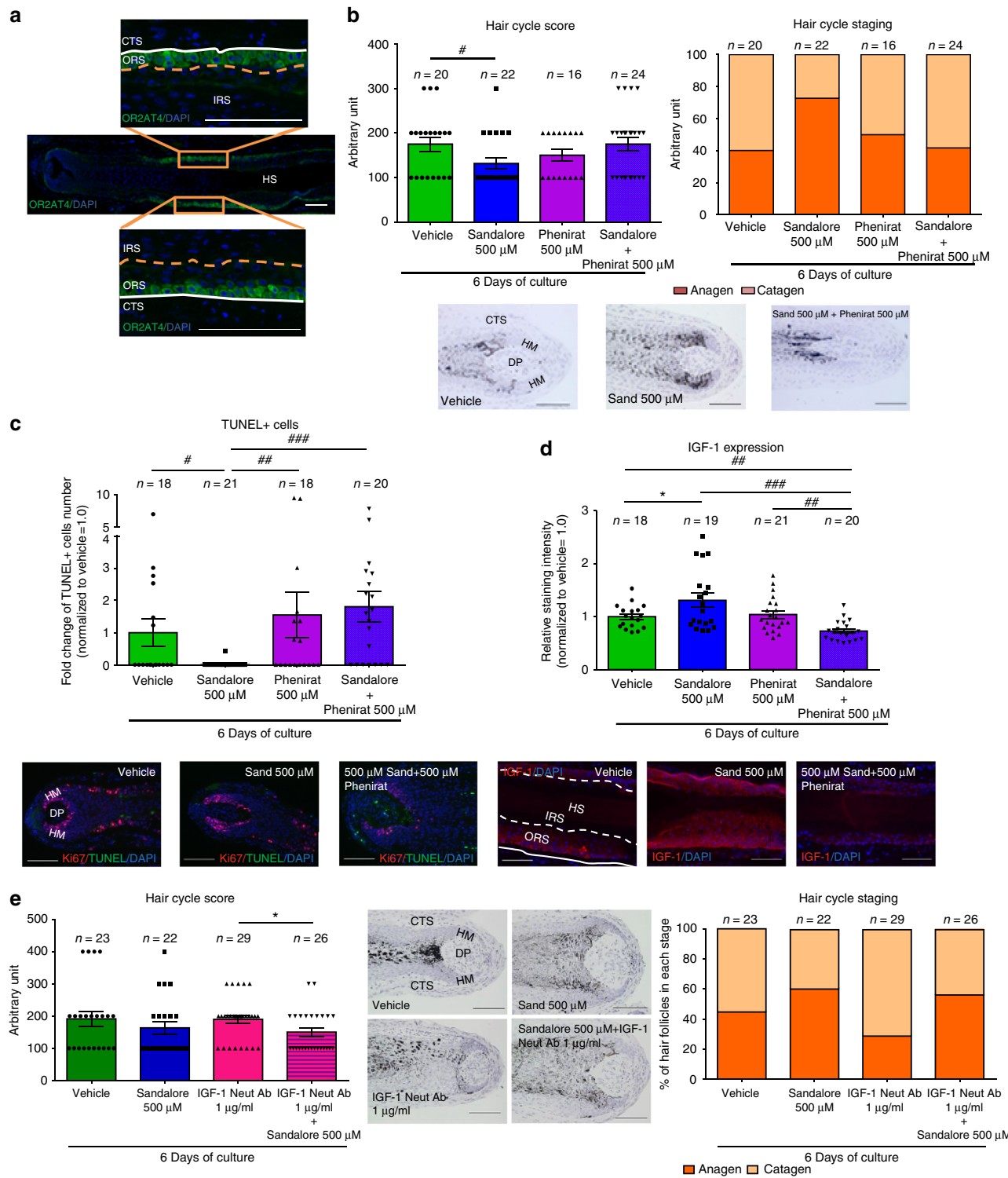

and Supplementary Data 1) are perfectly in line with anagen prolongation by Sandalore®.

Intriguingly, the strongest transcriptional upregulation (77.6× increase) was seen for *dermcidin*, a potent antimicrobial peptide with broad bactericidal activities that reportedly is only produced by sweat gland epithelium in human skin[40,41]. However, quantitative immunohistomorphometry confirmed that dermcidin protein is also upregulated by Sandalore® in the epithelium of human scalp HFs (Supplementary Fig. 7a and Table 1). This

demonstrates that human HFs also express dermcidin and raises the fascinating question whether OR2AT4 may act as a chemosensory receptor for selected bacterial metabolites, in response to which intrafollicular dermcidin production may be upregulated to manage the complex HF-microbiome[42,43].

When selected signaling pathways recognized to be involved in OR-mediated signaling[1,9,44–47], were studied by phospho-kinase assay[9], Sandalore® upregulated several expected kinase activities (Fig. 7a and Table 1). In line with our previous results (Fig. 1c, d),

**Fig. 1** Hair follicles express OR2AT4, which specific stimulation endorses IGF-1-dependent anagen prolongation. **a** Representative images showing OR2AT4 protein expression (using the previously published OR2AT4 antibody[20]) in proximal outer root sheath and hair matrix keratinocytes of human scalp microdissected hair follicles. **b** Hair cycle score and staging were evaluated in treated and vehicle HFs after 6 days of culture using Ki-67/TUNEL immunofluorescence and Masson–Fontana histochemistry[26]. Mean ± SEM, $n = 16$–24 HFs from three donors (independent experiments), Kruskal–Wallis test ($P = 0.0923$, n.s not significant) and Dunn's multiple comparisons test as post hoc test, ns not significant, Mann–Whitney test, $*P < 0.05$. Representative pictures of Masson–Fontana histochemistry in vehicle and treated HFs after 6 days of treatment. **c** Apoptotic hair matrix keratinocytes were counted in the hair matrix of all treated and vehicle HFs. Representative pictures of Ki67/TUNEL. Mean ± SEM, $n = 18$–21 HFs from three donors (independent experiments), Kruskal–Wallis ($P = 0.005$) test and Dunn's multiple comparisons test as post hoc test, $\#P < 0.05$, $\#\#P < 0.01$, $\#\#\#P < 0.001$. **d** IGF-1 expression was measured in ORS keratinocytes in treated and vehicle HFs. Representative pictures of IGF-1 immunofluorescence. IGF-1 expression was quantified in ORS keratinocytes in treated and vehicle HFs using ImageJ. Mean ± SEM, $n = 18$–21 HFs from three donors (independent experiments), Kruskal–Wallis ($P < 0.001$) and Dunn's multiple comparisons test as post hoc test, $\#\#P < 0.01$, $\#\#\#P < 0.001$, and Student's $t$-test, $*P < 0.05$. **e** Hair cycle score and staging were measured in treated and vehicle HFs after 6 days of culture. Representative pictures of vehicle and treated HFs after 6 days of treatment. Mean ± SEM, $n = 22$–29 HFs from three donors (independent experiments), Kruskal–Wallis test ($P = 0.1434$) and Dunn's multiple comparisons test as post hoc test, n.s significant, and Student's $t$-test after performing an iterative Grubbs outlier test, $*P < 0.05$. CTS connective tissue sheath, DP dermal papilla, HM hair matrix, ORS outer root sheath, IRS inner root sheath, HS hair shaft. Scale bar: 100 μm

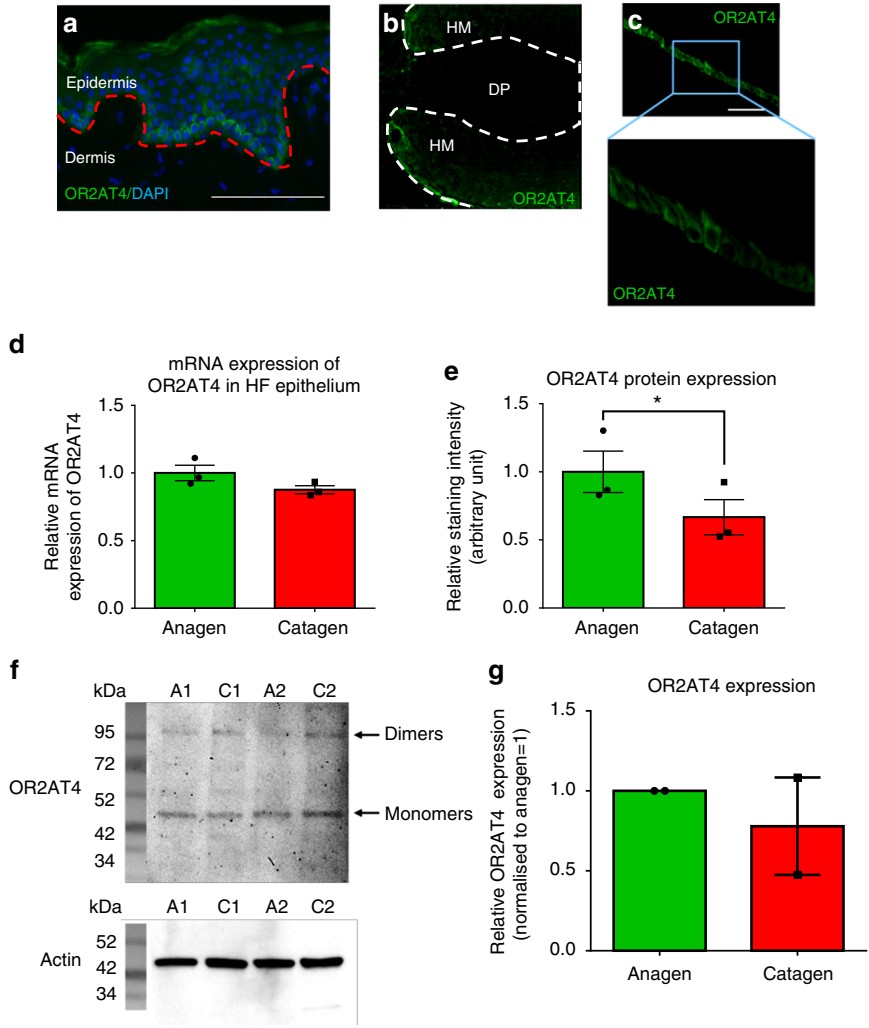

**Fig. 2** OR2AT4 mRNA and protein expression in human scalp epidermis and hair follicles in situ. **a** Representative pictures of OR2AT4 immunofluorescence in human scalp epidermis of three different donors (positive control[20]). Red line delineates the dermo-epidermal basement membrane. Cytosolic expression of OR2AT4 in hair matrix and suprabulbar outer root sheath (ORS) keratinocytes. Scale bar: 100 μm. **b**, **c** Representative pictures of confocal imaging of OR2AT4 immunofluorescence in human scalp HFs from three different donors (independent experiments). Scale bar: 100 μm. **d**, **e** mRNA (normalized against GAPDH) and in situ protein expression of OR2AT4 in anagen and catagen microdissected HF epithelium. Mean ± SEM, $n = 3$ from nine HFs/donor from three donors (independent experiments), Student's $t$-test, $*P < 0.05$. **f**, **g** Western blot analysis and quantitative results of OR2AT4 (normalized against actin) in anagen and catagen microdissected human scalp HFs. Mean ± SEM from nine HFs/donor from two donors (independent experiments). The specific band for OR2AT4 is found around 44 kDa, although the predicted molecular weight of OR2AT4 is 36 kDa. This slight difference can be explained by a post-translational modification (an acetylation site has been identified on the lysine at the position 303 [source: phosphoSitePlus®]) that would increase the molecular weight[64]. A1-2 indicates anagen HFs and C1-2 indicates catagen HFs from donor 1 and 2. DP dermal papilla, HM hair matrix

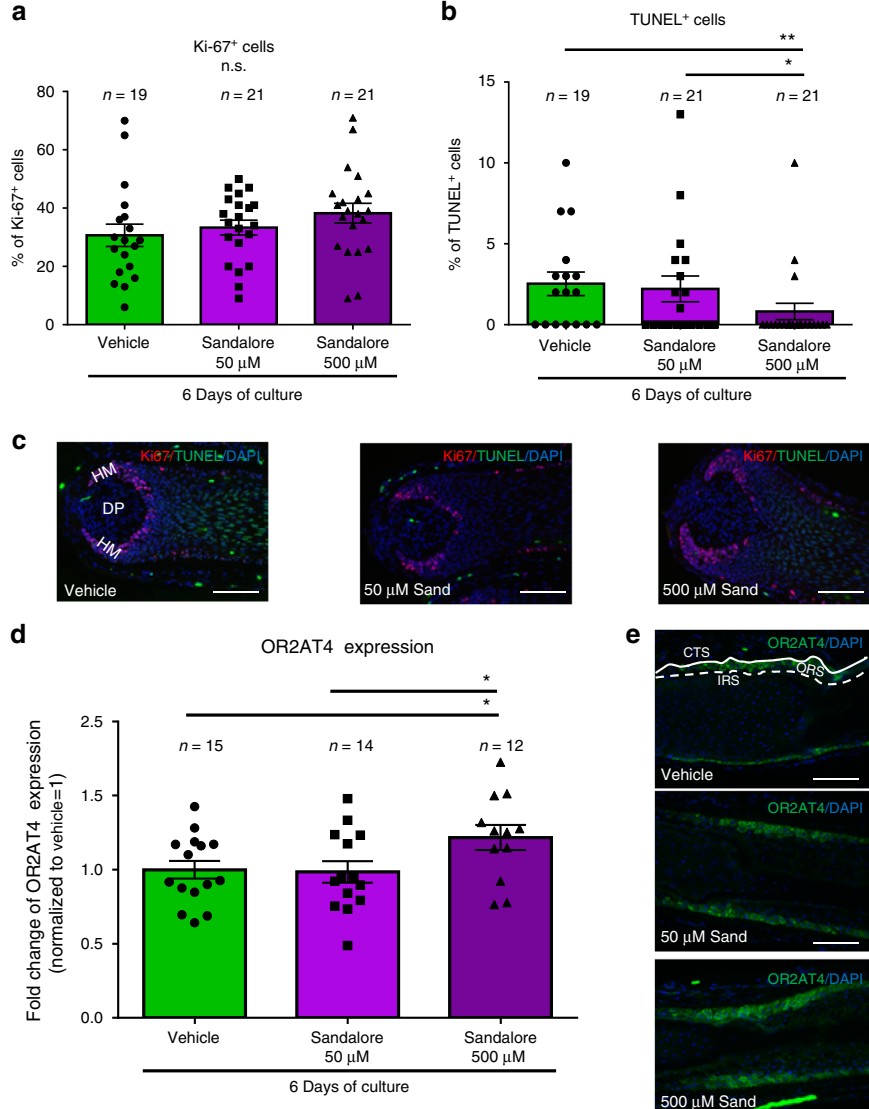

**Fig. 3** High concentration of Sandalore® (500 μM) regulates hair matrix keratinocytes apoptosis and intrafollicular OR2AT4 expression. **a**, **b** The number of Ki-67+ and TUNEL+ cells in the hair matrix was evaluated in the hair bulb of all treated and vehicle HFs. **c** Representative pictures of Ki-67/TUNEL staining. Mean ± SEM, n = 19–21 HFs from two donors (independent experiments), unpaired Student's t-test or Mann–Whitney test, *P < 0.05, **P < 0.01. DP dermal papilla, HM hair matrix. Scale bar: 100 μm. **d** OR2AT4 protein expression was evaluated using ImageJ in the ORS of Sandalore®-treated and control HFs after 6 days of culture. **e** Representative pictures of OR2AT4 expression in the ORS of cultured HFs. Mean ± SEM, n = 12–15 HFs from two donors (independent experiments), Student's t-test, *P < 0.05. CTS connective tissue sheath, IRS inner root sheath, ORS outer root sheath. Scale bar: 100 μm

this included increased phosphorylation of proline-rich AKT1 substrate 40 (PRAS40), whose expression is induced by IGF-1[48,49], while Sandalore® reduced phosphorylation of p53 (S46), which is highly phosphorylated in apoptotic cells[50] (IGF-1 is a key apoptosis suppressor[51–53]).

**Anagen-prolonging effect of Sandalore® implicates IGF-1**. Therefore, we next probed the hypothesis that, mechanistically, Sandalore® stimulation of OR2AT4 may retard catagen and suppress HF apoptosis by upregulating intrafollicular IGF-1-mediated signaling. Indeed, the co-administration of IGF1-neutralizing antibody with Sandalore® significantly reversed the catagen-promoting effect of IGF-1 neutralizing antibody alone (Fig. 1e; for extended discussion, see Supplementary Note 3). Mechanistically, this suggests that OR2AT4 activation mainly

prolongs anagen via upregulating IGF-1 expression and secretion by OR2AT4+ keratinocytes in the proximal ORS (Figs. 1d, 8). While IGF-1 signaling is known to be involved in olfactory bulb development and function[54–56], the current study reveals that IGF-1 expression/secretion in human epithelial tissue is also controlled by OR-mediated signaling and demonstrates that IGF-1 production underlies an OR2AT4-controlled chemosensory regulation.

## Discussion
Collectively, these data show that the growth, cyclic transformation, epithelial cell apoptosis, and IGF-1 production of a dynamic human (mini-)organ, i.e., scalp HFs[29], underlies an OR-dependent chemosensory control. Thus, human HFs can "smell" in the sense that they recruit the evolutionarily oldest and

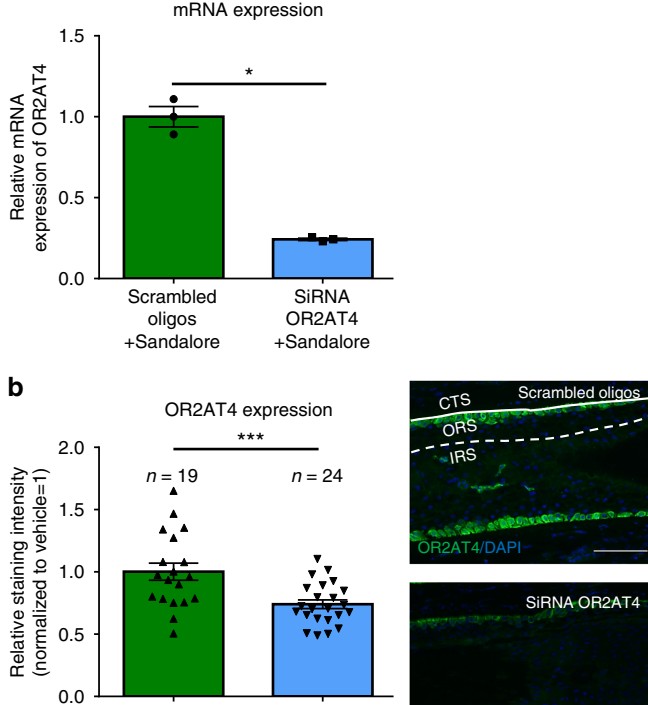

**Fig. 4** OR2AT4 expression in Scambled oligos or siRNA OR2AT4 Sandalore®-treated, microdissected human scalp HFs after 6 h of organ culture. **a** Microdissected, human scalp HFs were treated for 6 h with OR2AT4 siRNA or scrambled oligos, using the intrafollicular gene silencing technique[32]. OR2AT4 mRNA expression was evaluated after 24 h of culture in siRNA or scrambled oligo-transfected HFs. **b** OR2AT4 protein expression was evaluated using ImageJ in the ORS of siRNA-treated and control HFs after 6 days of culture. Representative pictures of OR2AT4 expression in the ORS of cultured HFs. Mean ± SEM, $n = 19$–24 HFs from three donors (independent experiments), Student's $t$-test, *$P < 0.05$, ***$P < 0.001$. CTS connective tissue sheath, IRS inner root sheath, ORS outer root sheath. Scale bar: 100 μm

largest of all receptor families[1,2] for regulating key organ functions (for extended discussion, see Supplementary Discussion 1). Moreover, we identify one specific OR, namely, OR2AT4, whose stimulation with a synthetic agonist (Sandalore®)[20] and whose selective silencing profoundly impacts on human hair growth ex vivo primarily via regulating expression and secretion of the key hair growth-promoting factor, IGF-1 (Fig. 8 and Supplementary Discussion 2). However, while IGF-1-mediated signaling is required for human hair growth promotion by Sandalore® (Fig. 1e), our phospho-kinase activity and gene expression profiling results suggest that additional pathways (e.g., p38a/ERK1/2/MSK1/2, HB-EGF/EGF-R, and FGF-7 pathways (Fig. 6, Supplementary Figs. 5, 6, and Supplementary Data 1)) are involved that deserve further exploration[57–61].

Perhaps most intriguingly, our silencing data suggest that OR2AT4-mediated signaling is required for maintaining human scalp HFs in anagen and for suppressing keratinocyte apoptosis in the hair matrix (Fig. 5d, e). This begs the question: What are the endogenous intrafollicular OR2AT4 ligands in human HFs? The endogenous ligands for human ORs remain to be definitively clarified, and those for OR2AT4 are unknown. Candidates

include molecules with Sandalore®-like structure, short-chain fatty acids[13], and—namely, in view of our dermcidin results (Supplementary Fig. 7a and Table 1)—metabolites of resident HF microbiota[42,43].

Taken together, our ex vivo data suggest that olfactotherapy by topically applied cosmetic OR2AT4 ligands like Sandalore® may promote human hair growth by prolonging anagen and inhibiting premature catagen development (e.g., in androgenetic alopecia and telogen effluvium).

Thus, using scalp HFs as accessible and tractable model organs and by selectively targeting OR2AT4, our study reveals an important, translationally relevant frontier in the OR-dependent chemosensory physiology of peripheral human tissues.

## Methods

**Human samples.** Temporal and occipital human scalp skin was obtained from healthy donors (38–69 years old) undergoing routine face-lift surgery after informed consent and ethical approval (University of Muenster, no. 2015-602-f-S). No sample size calculation was performed. Number of three different donors was used due to the small availability of the tissue used in the study. This number of three was used in many previous studies, given statistical significance.

**Tissue specimens.** Scalp skin samples were either cut into small pieces (4 mm), embedded into OCT, and frozen in liquid nitrogen, or processed for HF microdissection[25].

**HF organ culture.** Human scalp samples were obtained 1 day after face-lifting procedure (i.e., after overnight transport from collaborating surgeons) and used at the same day for microdissecting human anagen VI scalp HFs. The HF microdissection technique employed for setting up the classical Philpott assay[25,26,62] used in the current study, removes all perifollicular tissue with the sole exception of the HF's dermal sheath, and thus does not contain any other skin appendage structures (e.g., eccrine gland elements)[25]. Microdissected human scalp HFs were cultured at 37 °C with 5% $CO_2$ in a minimal media of William's E media (WEM, Gibco, Life Technologies) supplemented with 2 mM of L-glutamine (Gibco), 10 ng/ml hydrocortisone (Sigma-Aldrich), 10 μg/ml insulin (Sigma-Aldrich), and 1% penicillin/streptomycin mix (Gibco)[25,26,62]. After microdissection, the HFs were first incubated in WEM for 24 h for re-equilibration. HFs after quality control (fully pigmented and presence in anagen VI phase) were randomly allocated to the different experimental groups.

**Chemical stimulation of human microdissected HFs.** After 24 h, WEM medium was replaced and HFs were treated with vehicle (0.1% DMSO), Sandalore® (50 and 500 μM; see Fig. 3 and Supplementary Note 1, Symrise), Phenirat® (in a ratio 1:1 to the agonist, Symrise), or Sandalore®+Phenirat® for 6 days for (immuno-)histology or 6 h for qRT-PCR.

For the IGF-1 neutralizing antibody experiments, IGF-1 neutralizing antibody (1 μg/ml, ab9572, Abcam) was added 30 min before adding Sandalore® to the corresponding groups. Culture medium was replaced every second day and after 6 days. HFs were then embedded in cryomatrix (Fisher Scientific), and snap frozen in liquid nitrogen for (immuno-)histology.

**SiRNA transfection-knockdown OR2AT4 in organ-cultured HFs.** Human anagen VI HFs were transfected using a commercial siRNA reagent system (Sc-45064, Santa Cruz) following the manufacturer's instructions[32,34,35]. Briefly, stock solutions (10 μM) of siRNA OR2AT4 (gift from Prof. Hanns Hatt[20]) and siRNA control (scrambled oligo) were prepared using RNAse-free water. HF transfection was performed 24 h after microdissection for 6 h using either 100 mM OR2AT4 siRNA or control scramble siRNA. After 24 h of incubation with fresh WEM medium, HFs were collected per group in RNA later and stored at 4 °C for further RNA extraction and qRT-PCR analysis or immediately frozen in liquid nitrogen and stored at −80 °C for microarray analysis. Finally, fresh WEM medium was replaced every second day and after 5 days of culture, HFs were snap frozen in OCT for further quantitative (immuno-)histomorphometry analysis.

**Histology.** For histochemical visualization of melanin, Masson–Fontana staining was performed on frozen sections. Melanin was stained as brown dots[26].

**Immunofluorescence.** OCT-embedded samples were sectioned (6 μm thickness for HF and 7 μm thickness for skin) with a Leica cryostat. For primary OR2AT4[20] (custom designed rabbit polyclonal antibody generated against the C-terminus sequence of OR2AT4 (Eurogentec, Liège, Belgium)), or cleaved-caspase-3 (#9661, clone Asp175, Cell Signaling) antibodies staining, tissue cryosections were fixed in 4% paraformaldehyde, pre-incubated with 10% of goat serum (for OR2AT4) or 5% goat serum +0.3% Tritton X-100 (for cleaved-caspase 3) and incubated with the

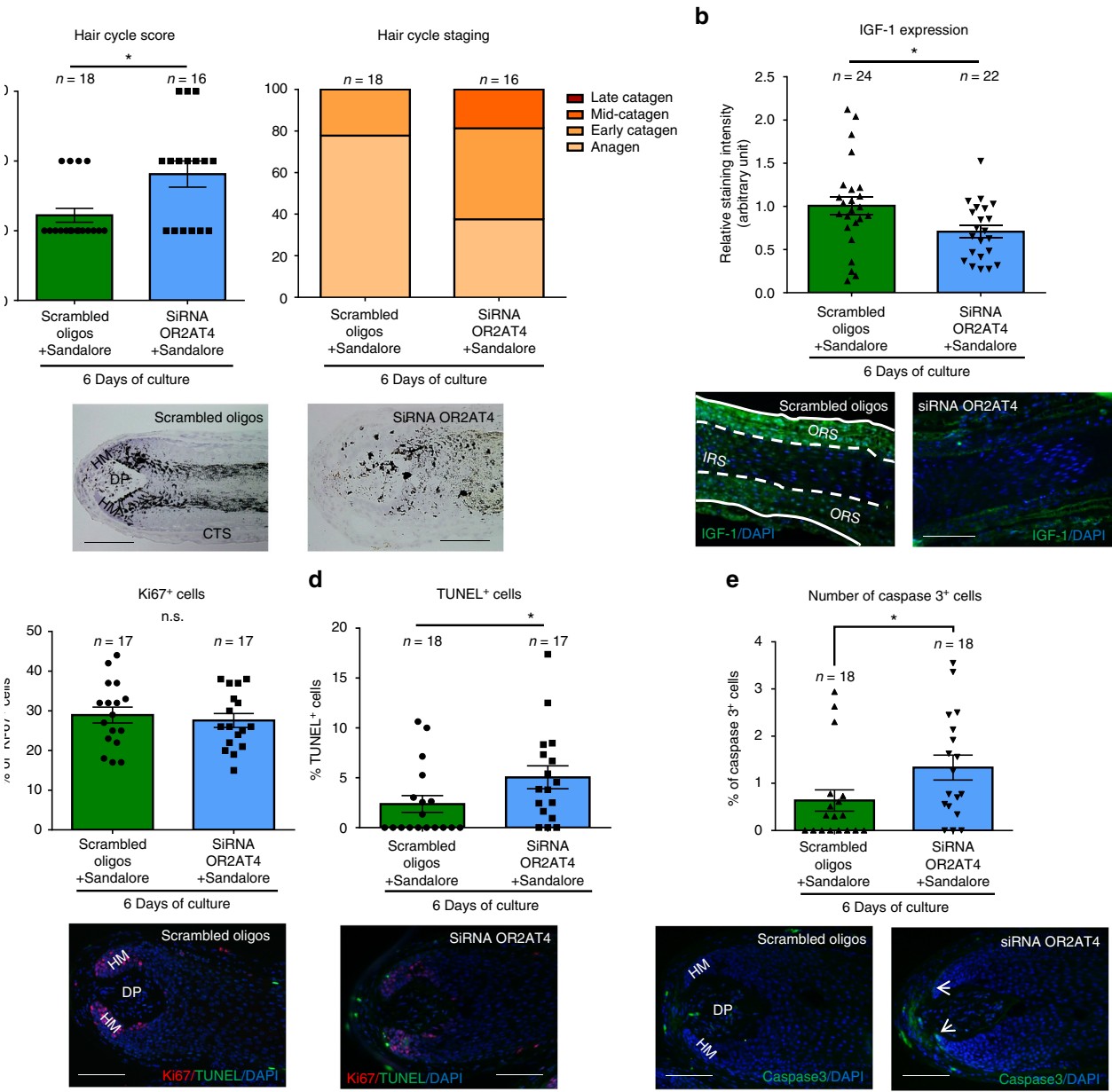

**Fig. 5** The anagen prolongation effect of Sandalore® is OR2AT4 dependent. **a** Hair follicle cycle score and staging were performed in HFs treated with siRNA OR2AT4 or scrambled oligos in the presence of Sandalore®[32]. Representative images of vehicle and treated HFs after 6 days of treatment. Mean ± SEM, n = 16–18 HFs from three donors (independent experiments), Mann–Whitney test, *P < 0.05. **b** IGF-1 expression was quantified in ORS keratinocytes in treated and vehicle HFs using ImageJ. Representative pictures of IGF-1 immunofluorescence. Mean ± SEM, n = 22–24 HFs from three donors (independent experiments), Mann–Whitney test, *P < 0.05. **c**, **d** Ki67+ cells and TUNEL+ cells were counted in the hair matrix of siRNA and control HFs. Representative pictures of Ki67/TUNEL double-staining in the hair bulb of HFs. Mean ± SEM, n = 17–18 HFs from three donors (independent experiments), Mann–Whitney test, *P < 0.05, n.s. not significant. **e** The number of cleaved caspase-3+ cells (white arrows) in the hair matrix was evaluated in the hair bulb of all HFs treated with OR2AT4-siRNA or scrambled oligos. Representative pictures of cleaved caspase-3 staining. Mean ± SEM, n = 18 HFs from three donors (independent experiments), Student's t-test, *P < 0.05. CTS connective tissue sheath, DP dermal papilla, HM hair matrix, ORS outer root sheath, IRS inner root sheath. Scale bar: 100 μm

corresponding primary antibody at 4 °C overnight (1/100 for OR2AT4 and 1/400 for cleaved-caspase 3). Secondary antibody incubation was performed at RT for 45 min. Counterstaining with DAPI (1 μg/ml) was performed to visualize nuclei.

Dermicidin protein was detected using tissue sections fixed in 4% paraformaldehyde, pre-incubated with 10% of goat serum, and incubated with a mouse anti-human Dermcidin antibody (Novus Biologicals, G-81, 1:200). Secondary antibody (Goat anti-mouse Alexa fluor 488) incubation was performed

at room temperature for 45 min. Counterstaining with DAPI (1 μg/ml) was performed to visualize nuclei.

For TGFβ2 (Sc-90, Santa Cruz) and IGF-1 (Sc-1422, clone G-17, Santa Cruz[31,32]), tissue cryosections were fixed in acetone and endogenous peroxidase activity was blocked with 3% of H2O2 (Merck Milipore). This step was followed by an avidin-biotin blocking step (SP2001, Vectorlabs) and a preincubation with TNB buffer (Tris HCl+NaCl+Casein). The corresponding primary antibody was

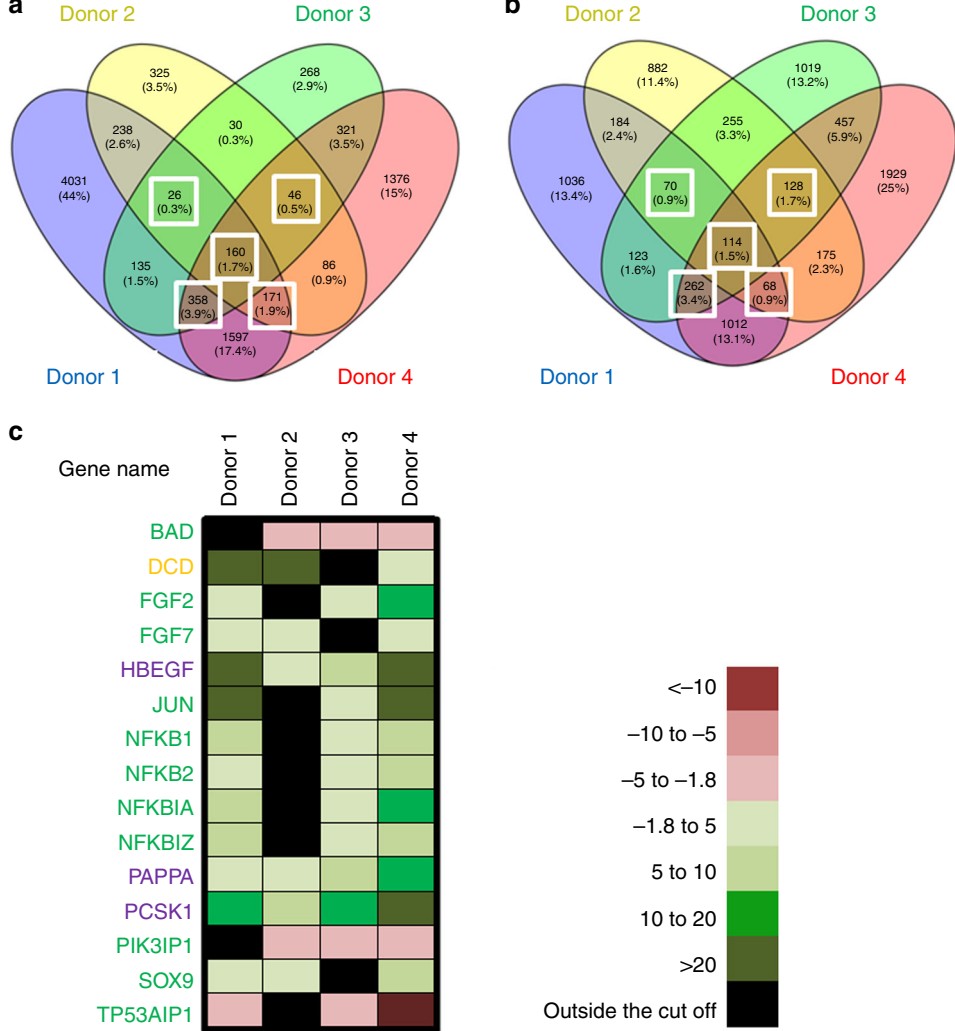

**Fig. 6** Microarray-based analysis of genes related to anagen-prolonging pathways after stimulation with Sandalore® (500 μM). **a**, **b** Venny diagrams[65] show the upregulated and downregulated genes (cut-off: fold change >−1.8 or >+1.8 and equidirectional changes). White squares indicate genes upregulated and downregulated in at least three of four donors (independent experiments). The heatmap shows the list and the expression level of the most upregulated and downregulated genes related to the different pathways regulated after OR2AT4 activation (cut-off: fold change >−1.8 or >+1.8 and equidirectional changes) in at least three of four donors (independent experiments) (**c**). Green: apoptosis related, orange: dermcidin related, and violet: IGF related

incubated at 4 °C overnight (1/1000 for TGFβ2 and 1/250 for IGF-1). Secondary antibody incubation was performed at RT for 45 min before using the Tyramide signal amplification kit (NEL700001KT, Perkin Elmer). Counterstaining with DAPI was performed to visualize nuclei.

To stain apoptotic and proliferating cells, we used the apoptag kit (Merck Milipore) following the manufacturer's protocol followed by Ki-67 staining[25,26,33,63]. Primary antibody was incubated overnight (Ki-67, M7240 Clone: MIB-1, DAKO, 1/20) after the TdT-enzyme step. The secondary antibody was incubated for 45 min at RT after the fluorescent-labeled anti-Digoxigenin step of the apoptag kit. Counterstaining with DAPI was performed to visualize nuclei. Negative controls were performed by omitting the primary antibody. Images were taken using a Keyence fluorescence microscope BZ9100 (Osaka, Japan) maintaining a constant set exposure time throughout imaging for further analysis.

**Quantitative reverse transcriptase-PCR**. Total RNA was isolated from whole microdissected HFs using RNeasy Mini Kit (Quiagen) following the manufacturer's instructions described in the manufacturer's protocol. RNA purity and concentrations were determined using the Nanodrop ND-1000 assay (Fisher Scientific). Reverse transcription of the RNA into cDNA was performed using the TetrocDNA Synthesis Kit (Bioline), according to the manufacturer's instructions. RNA concentrations were adjusted between 50 to 500 nM for each sample set to allow further quantification comparison between samples and experiments after qRT-PCR. Controls were performed using the housekeeping gene *GAPDH*. Real-

time quantitative polymerase chain reaction (qRT-PCR) was run in triplicate using TaqMan Fast Advanced Master Mix Product Insert and gene Expression Assay transcripts (Id: Hs01060665_g1 for *ACTB*, Hs02758991_g1 for *GAPDH*, and Hs02339277_s1 for *OR2AT4*, Applied Biosystem) on the qTower2.2 thermocycler. Real-time quantification plots and Ct values were collected and stored by the qPCRsoft2.1 software. The amount of the transcripts was normalized to those of the housekeeping gene using the ΔΔCT method using EXCEL.

**Whole-genome microarray analysis**. RNA isolation, sample processing, and microarray analyses (Agilent Technologies), as well as statistical evaluation, were performed by Arrows Biomedical GmbH (Muenster, Germany). Expressional alteration was considered to be significant only when ≥1.8-fold and equidirectional changes were observed in at least three of four patients (independent experiments). An additional analysis has been performed using 5-fold and equidirectional changes in the four different donors (independent experiments) in order to identify the top up and downregulated genes.

**Human phospho-kinase array**. In order to gauge which signaling pathways are regulated by the specific stimulation of OR2AT4, we performed a phospho-kinase array[9]. Total protein was isolated from whole microdissected HFs using a specific buffer from the Human Phospho-Kinase Array (ARY003B, R&D System), following the manufacturer's protocol. Briefly, protein extracts were diluted and incubated overnight with the Human Phospho-Kinase Array. The array was

**Table 1 OR2AT4 stimulation by Sandalore® treatment regulates different signaling protein phosphorylation pathways in human HFs ex vivo**

| Proteins regulated under 50 μM Sandalore stimulation | Type of regulation | Proteins regulated under 500 μM Sandalore stimulation | Type of regulation |
|---|---|---|---|
| p38α (T180/Y182) | ↑(2,5) | p38α (T180/Y182) | ↑(1,8) |
| ERK1/2 (T202/Y204, T185/Y187) | ↑(1,9) | ERK1/2 (T202/Y204, T185/Y187) | ↑(1,8) |
| EGFR (Y1086) | ↑(3,2) | EGFR (Y1086) | ↑(2,6) |
| MSK1/2 (S376/S360) | ↑(2,1) | MSK1/2 (S376/S360) | ↑(2,1) |
| PYK2 (Y402) | ↑(4,7) | PYK2 (Y402) | ↑(2,2) |
| Hsp60 | ↓(0,6) | | |
| JNK 1/2/3 (T183/Y185, T221/Y223) | ↑(2,0) | | |
| AMPKα1 (T183) | ↑(2,0) | | |
| PLC-γ1 (Y783) | ↑(2,4) | | |
| | | Fgr (Y421) | ↑(1,9) |
| | | PRAS40 (T246) | ↑(1,9) |
| | | p53 (S392) | ↓(0,5) |
| | | p53 (S46) | ↓(0,6) |
| | | p53 (S15) | ↓(0,5) |

*PRAS40* proline-rich AKT1 substrate 40[49], *S46* phosphorylated p53[50], *p53* phosphoprotein p53, *p38α* mitogen-activated protein kinases 14, *ERK1/2* extracellular signal-regulated kinases1/2, *EGFR* epidermal growth factor receptor, *MSK1/2* mitogen and stress-activated protein kinase 1/2, *PYK2* protein tyrosine kinase 2, *Hsp60* heat shock protein 60, *JNK1/2/3* c-Jun N-terminal protein kinase 1/2/3, *AMPKα1* AMP-activated protein kinase α1, *PLCγ* phospholipase C γ1, *Fgr* Feline Gardner-Rasheed proto-oncogene, *WNK1* WNK lysine-deficient protein kinase 1 isoform
Protein extraction and phospho-kinase assay was performed in treated and vehicle human microdissected HFs following the manufacturer's protocol (for Supplementary Information, see Methods section). Summary of kinases upregulated at least (1.8×-fold change of the control) or downregulated by >50% of the control by Sandalore® (50 and 500 μM, respectively)

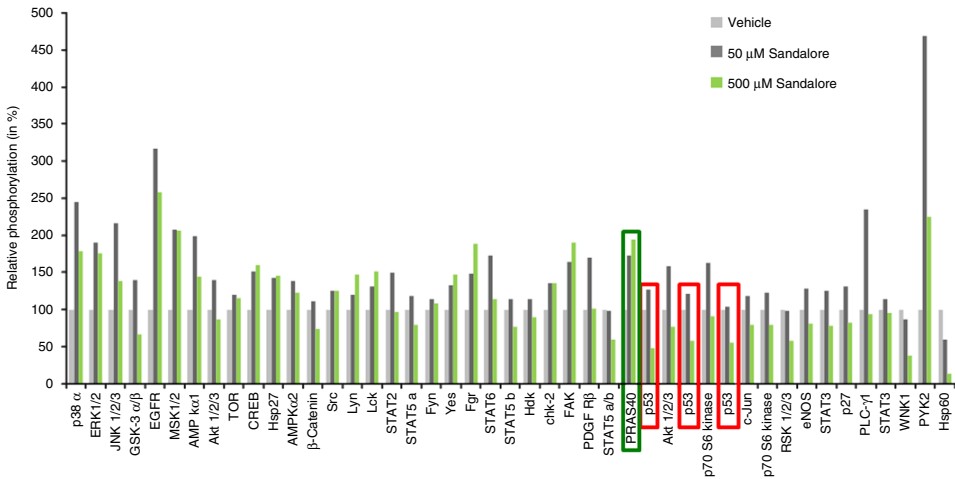

**Fig. 7** OR2AT4 stimulation by Sandalore® treatment regulates different protein phosphorylation signaling pathways in human HFs ex vivo. Plot showing regulation of the phosphorylation of 45 kinases mediated by Sandalore® (50 and 500 μM) in cultured microdissected HFs from four different donors (independent experiments). Among those kinases, the most interesting ones underline in green (upregulated) and red (downregulated) rectangles. PRAS40 proline-rich AKT1 substrate 40[49], S46 phosphorylated p53[50], p53 phosphoprotein p53, p38α mitogen-activated protein kinases 14, ERK1/2 extracellular signal-regulated kinases1/2, EGFR epidermal growth factor receptor, MSK1/2 mitogen- and stress-activated protein kinase 1/2, PYK2 protein tyrosine kinase 2, Hsp60 heat shock protein 60, JNK1/2/3 c-Jun N-Terminal Protein Kinase 1/2/3, AMPKα1 AMP-activated protein kinase α1, PLCγ phospholipase C γ1, Fgr Feline Gardner-Rasheed proto-oncogene, WNK1 WNK lysine-deficient protein kinase 1 isoform

washed to remove unbound proteins, followed by incubation with a cocktail of biotinylated detection antibodies. Streptavidin-HRP and chemiluminescent detection reagents were applied, and a signal was produced at each capture spot corresponding to the amount of phosphorylated protein bound.

**Western blot analysis**. Total protein was extracted from nine anagen and catagen microdissected human scalp HFs. Protein concentrations were determined using a Bradford assay (B6916, Sigma-Aldrich). Thirty micrograms of protein were subjected to 4–15% Mini-PROTEAN® TGX™ Precast gel (#4561083, Bio-Rad) and transferred to a nitrocellulose membrane (88018, Thermo Fisher Scientific), followed by incubation with the corresponding primary antibodies (PA5-71599 for OR2AT4, 1/1000, Thermo Fisher Scientific; and A3853 for Actin, 1/1000, Sigma-Aldrich) overnight at 4 °C. After incubation with peroxidase-conjugated secondary antibodies (WesternBreeze™ Chemiluminescent Kit, WB7106 and WB7104, Thermo Fisher Scientific), the bands were visualized using Chemocam imager 6.0

(Intas, Germany). Protein expression levels were normalized to corresponding actin levels. The uncropped blots are presented in the Supplementary Fig. 8.

**Hair cycle score (HCS) and staging**. HFs were microscopically evaluated for the hair cycle staging analysis using Masson–Fontana histochemistry and Ki-67/TUNEL immunostainings[25,26]. The HCS was also measured[25,31], which consists of assigning an arbitrary unit for each stage of the hair cycle (Anagen VI = 100; Early catagen = 200; Mid-catagen = 300; and Late catagen = 400). After having classified each HF according to its hair cycle stage, following the previously defined objective classification criteria for organ-cultured human HFs[26], for each experimental condition, the mean HCS was calculated. The closer the mean is to 100, the higher is the number of anagen VI HFs in a given group. The HCS provides a global read-out parameter that looks at all HFs in a given experimental group and synthesizes them into a single number, which reflects how close the majority of HFs is to either anagen VI or catagen and also permits statistical analysis that it is not possible with

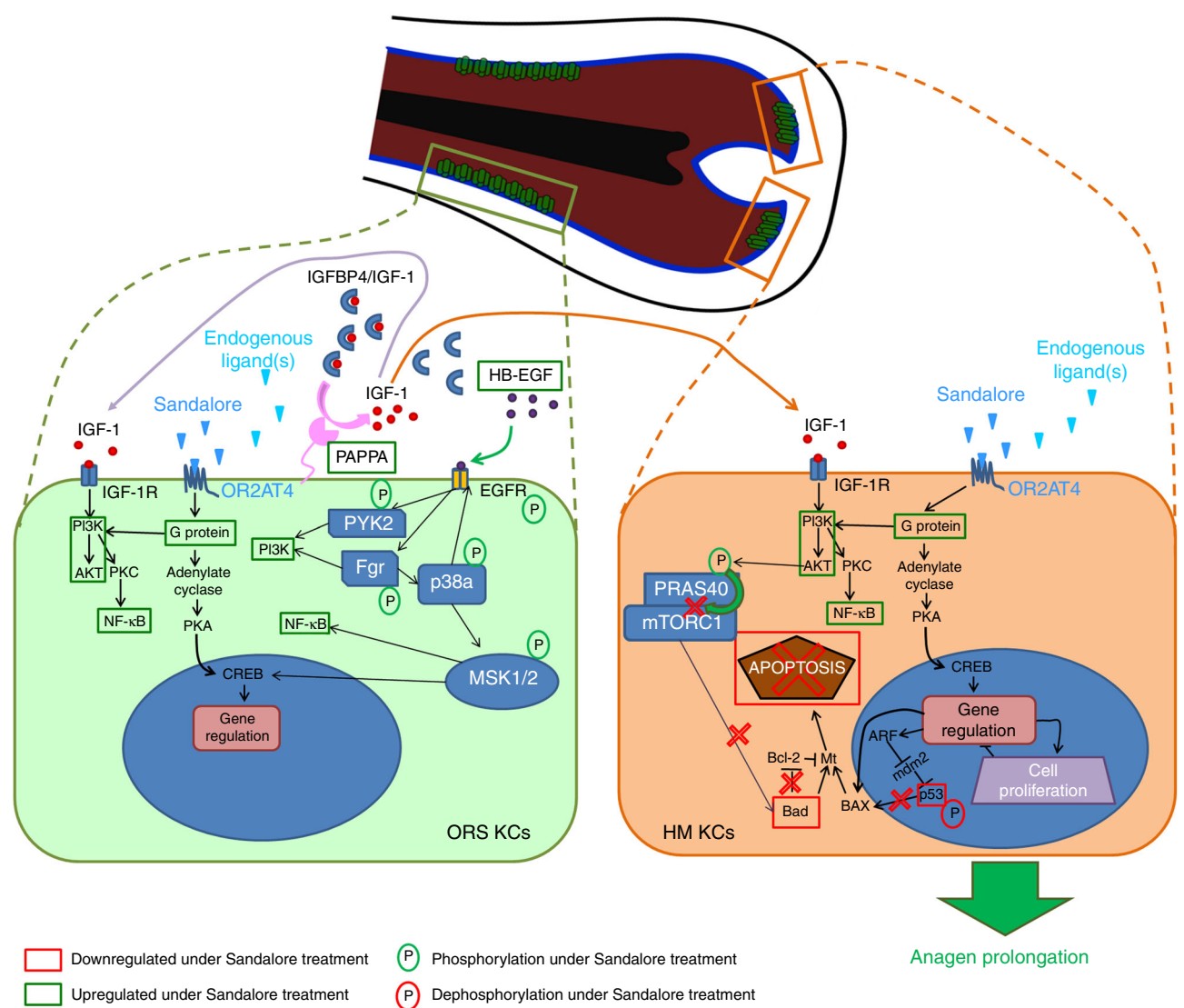

**Fig. 8** Proposed mechanism of action of OR2AT4 activation by Sandalore® and (unknown) endogenous ligand(s) in human hair follicle epithelium. The activation of OR2AT4 at the cell surface of outer root sheath keratinocytes (ORS KCs; location: see green cells in the central HF cartoon) by endogenous ligands and/or Sandalore® upregulates the expression of genes and kinases involved in programmed cell death, thus preventing intrafollicular apoptosis (e.g., by phosphorylation of PRAS40 preventing its interaction with mTOR1, upregulation of NF-κB pathway) or downregulates key players in the apoptotic machinery (e.g., dephosphorylation of p53, downregulation of Bad). In parallel, OR2AT4 activation by exogenous (Sandalore®) or endogenous ligands (e.g., metabolites of the HF microbiome) induces the upregulation of PAPPA that cleave the IGFBP4/IGF1 complex to release IGF-1 (pink arrows). The released IGF-1 triggers the activation of IGF-1R on the same cell (autocrine signaling, purple arrow) or on hair matrix keratinocytes (HM KCs; orange "cell") (paracrine signaling, orange arrow). The activation of IGF-Rs on HM keratinocytes then induces signaling cascades (e.g., PI3K/AKT and/or p38a/ERK1/2/MSK1/2) that activate different transcription factors and particularly CREB, which results in an anti-apoptotic effect and prolonged anagen phase in human HFs. P phosphorylation, green square gene upregulation, red square gene downregulation, green circle phosphorylation, red circle dephosphorylation

hair cycle staging. Therefore, hair cycle staging and the HCS are independent read-out parameters that complement each other.

**Quantitative (immuno-)histomorphometry.** Staining intensity was evaluated in well-defined reference areas by quantitative (immuno-)histomorphometry [31,32], using NIH ImageJ software (NIH, Bethesda, MD, USA).

**Statistical analyses.** All data are expressed as mean ± SEM (and variance is different between the groups) and were analyzed by one-way ANOVA or Kruskall–Wallis test and Dunn's multiple comparisons test as post hoc test when more than two groups were compared or Student's t-test or Mann–Whitney test when only two groups were compared (GraphPad Prism 6, GraphPad Software, San Diego, CA, USA) after performing d'Agostino and Pearson omnibus normality test. $P < 0.05$ was regarded as significant.

## Data availability

The data discussed in this publication are available from the authors and the microarray data have been deposited in NCBI's Gene Expression Omnibus (GEO) and are accessible through GEO Series accession number GSE102887.

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

## Acknowledgements

The excellent technical assistance of Janine Jakobs and Arrows Biomedical GmbH for generating the microarray data are gratefully acknowledged. We thank Dr. Christopher Ward for his collegial assistance and advice with microarray data mining, our collaborating surgeons (namely, Dr. Hanieh Erdmann, and Dr. Wolfgang Funk) for their invaluable cooperation, and Giammaria Giuliani for continued encouragement, stimulating discussions, and support. This study was supported in part by a research grant from Giuliani Pharma S.p.A., Milano, to Monasterium Laboratory GmbH, Münster.

## Author contributions

R.P. and H.H. conceived the study. R.P. designed and supervised the study, and obtained funding. J.C. performed most of the experiments, assisted with experimental design, and analyzed the data, assisted by M.B., L.P. J.L., and M.A. T.T. ran the phosphokinase assays. R.P., M.B., H.H. and J.C. interpreted the data. J.C., M.B. and R.P. wrote the manuscript, which was edited and approved by all authors.

## Additional information

**Competing interests:** J.C., M.B., L.P., J.L. and M.A. are or were employees of Monasterium Laboratory GmbH, Münster, which was founded by R.P. R.P., also serves as consultant for Giuliani Pharma, which has filed a patent on the use of compounds and compositions targeting OR2AT4 for hair growth-promotion or inhibition in humans (wo2017198818 (a1)—compounds for promoting hair growth and/or inhibiting or delaying hair loss in humans, and compositions for such uses). The remaining authors declare no competing interests.

