## [Peer Review File · Nature Communications]

Reviewers' comments:

Reviewer #1 - expert in hair follicles (Remarks to the Author):

In this manuscript Cheret et al. report a novel function of the olfactory receptor OR2AT4 in the regulation of the hair growth in an ex vivo system. OR2AT4 whose endogenous ligand is unknown, was previously shown to be expressed and have a functional role in human epidermis and wound healing. Furthermore, OR2AT4 can be pharmacologically modulated with a synthetic ligand (Sandalore) or antagonist (Phenirat), which is exploited by the authors in the current study. Specifically, using these reagents to treat human hair follicles (HFs) ex vivo they report that Sandalore prolongs hair growth and retards apoptosis compared to control. Microarray analysis of Sandalore-treated HFs showed that upregulation of IGF-1 as well as other pathways involving various kinases. Knock-down of OR2AT4 reversed attenuated these effects in the ex vivo assay suggesting the sustained hair growth is mediated by OR2AT4 through IGF-1.

Based on these data the authors discuss the possibility of pharmacologically exploiting HF-specific olfactory receptors to treat various hair disorders, which is of great clinical importance and has the potential to capture the attention of the general audience. For exactly these reasons I believe that several of the claims and conclusions that the authors make in this manuscript deserve to be further supported by additional experimental evidence.

My major concern is that all the conclusions from this study are based on a single ex vivo assay, of human hair follicle growth on a dish. While this is a very valuable assay in the experimental toolbox of a skin and hair biologists, it is not without caveats. Furthermore, many of the results that the authors present from this assay while statistically significant are still quite modest and require further validation with alternative in vitro or in vivo approaches. The authors mention at the beginning of the manuscript that olfactory receptors are evolutionarily ancient chemosensory systems. To my knowledge, mice have a OR2AT4 homolog which may also have a conserved role in HF growth. The authors should at least discuss the possibility of studying these chemoreceptors in vivo. Also, as a synthetic fragrance Sandalore may already be used on skin care products. One can envision a clinical study to assess the effects of such formulations to hair growth.

Other comments:

1) The authors need to explain the "Hair cycle score" better because despite the citation it is difficult to follow for the non-hair expert.

2) It is not clear to me why treatment with the antagonist Phenirat alone does not retard hair growth further compared to vehicle. The idea is that under baseline conditions hair follicles ex vivo will grow to some extent and that growth is in part sustained by a still active OR2AT4. Therefore, treatment with the antagonist should impair that baseline growth.

3) The differences seen in Fig1e appear to be non-significant

4) The phosphokinase assay suggests that the effects of Sandalore treatments are not always dose-dependent. The authors should comment on this and also consider re-evaluating some of their experiments taking the dose into consideration. It is not clear why 500µm was chosen for all the experiments.

5) The authors discuss the possibility of microbial involvement in hair physiology through OR2AT4, based on the finding dermcidin shows the strongest up-regulation following treatment with Sandalore. Perhaps this result is due to cross-contamination from sweat glands that were not fully dissociated from the HF during the microdissections. The authors have previously reported on the integral relationship between sweat glands and HFs so this is not a remote possibility.

6) The authors perhaps could take advantage of the OR2AT4 antibody to more accurately characterize expression levels and cellular specificity during the different stages of the hair cycle by western blot analysis and flow cytometry. For example in their RNAi experiments a western blot would give a more accurate view than immunofluorescence.

Reviewer #2 - expert in olfactory receptors (Remarks to the Author):

The overall findings of this paper are quite interesting. There is a growing awareness that olfactory receptors can be found in numerous other tissues (the authors are not comprehensive in their citations of this) and that they have in many cases unknown functions and unknown ligands. The Sandalore receptor is a case in point. This receptor is not activated for example by natural sandalwood but only by the synthetic Sandalore. Its endogenous ligand in HFs is completely unknown.

The authors use a variety of controls to show that the effects are produced by the activation of the Sandalore OR, including antagonists, knockdown and antibody blockers. In general it appears that treatment with Sandalore inhibits apoptosis in HF cells, although it is not clear which of several pathways are involved – or perhaps all of them. There is some concern that the effects of sandalore require exposure to relatively high concentrations for 6 days. Under those circumstances can the authors confidently rule out non-specific effects, possibly even membrane permeability to this volatile compound, that may be at work?

The possibility of a hair loss treatment is enticing but the data on p53 pathway inhibition and the blocking of apoptosis suggests a potential danger of carcinogenic effects.

This paper is a useful contribution to the growing literature on olfactory receptor expression in non-nasal tissue. My only objection in this regard is that there seems to me to be some loose language in this paper that confuses smelling or olfactory processes with chemical detection, which is a general cellular function.

For example in the abstract, and elsewhere, the authors state that human HFs can “smell” ... putting the word smell in quotes does nothing to clear up the confusion.

Response to reviewer's comments:

Reviewer #1

In this manuscript Cheret et al. report a novel function of the olfactory receptor OR2AT4 in the regulation of the hair growth in an ex vivo system. OR2AT4, whose endogenous ligand is unknown, was previously shown to be expressed and have a functional role in human epidermis and wound healing. Furthermore, OR2AT4 can be pharmacologically modulated with a synthetic ligand (Sandalore[®]) or antagonist (Phenirat[®]), which is exploited by the authors in the current study. Specifically, using these reagents to treat human hair follicles (HFs) ex vivo they report that Sandalore[®] prolongs hair growth and retards apoptosis compared to control. Microarray analysis of Sandalore[®]-treated HFs showed that upregulation of IGF-1 as well as other pathways involving various kinases. Knock-down of OR2AT4 reversed attenuated these effects in the ex vivo assay suggesting the sustained hair growth is mediated by OR2AT4 through IGF-1.

Based on these data the authors discuss the possibility of pharmacologically exploiting HF-specific olfactory receptors to treat various hair disorders, which is of great clinical importance and has the potential to capture the attention of the general audience. For exactly these reasons I believe that several of the claims and conclusions that the authors make in this manuscript deserve to be further supported by additional experimental evidence.

My major concern is that all the conclusions from this study are based on a single ex vivo assay, of human hair follicle growth on a dish. While this is a very valuable assay in the experimental toolbox of a skin and hair biologists, it is not without caveats.

The reviewer may not be entirely aware of the fact that this assay, first introduced in 1990 (Philpott et al., 1990), and refined by us and many other colleagues (see Langan et al., 2015), after decades of having been employed in numerous scholarly studies, is now accepted world-wide as the most clinically relevant of all preclinical assays in human hair research and arguably has the highest predictive value for the clinical effects that any agent tested in it will show in subsequent clinical trials. This unique assay is rivalled only (and complemented) by the xenotransplantation of human scalp skin/hair follicles (HFs) onto immunocompromised mice (see Oh et al., 2016).

Given that we have probably published more papers using this assay than any other group, we are well aware of its limitations and have critically reviewed these (Langan et al., 2015). However, for addressing exactly the key questions that matter most here, namely whether or not OR2AT4 stimulation and OR2AT4 silencing manipulate the growth phase (anagen) of human scalp HFs and for dissecting at least one plausible mechanism of action for any hair growth-modulatory effect mediated by OR2AT4, this assay is not only perfectly sufficient, but also better-suited than any other method currently available in preclinical human hair research. Therefore, it remains unclear which specific caveats the reviewer is thinking of here.

Furthermore, many of the results that the authors present from this assay while statistically significant are still quite modest and require further validation with alternative in vitro or in vivo approaches.

Here, we respectfully disagree. The differences between vehicle controls, Sandalore[®]-treated or OR2AT4-silenced HFs with respect to anagen duration, hair matrix apoptosis, and IGF-1 protein expression are not only significant, but also substantial in comparison to those routinely reported by us and independent groups with other test agents in this assay. If the effects seen here within 6 hrs (for mRNA analysis) or 6 days (for protein analysis) of treatment *ex vivo* can be extrapolated to the *in vivo* situation, where Sandalore[®] would be administered for many weeks or even months, it is reasonable to assume that the substantial effects already observed during the narrow window of observation *ex vivo* will be even more striking after long-term application *in vivo*.

Also, the fact alone that we conclusively demonstrate here for the very first time that any odorant, and the knock-down of any olfactory receptor, greatly impacts on a) the spontaneous remodelling of a human (mini-organ) and b) the protein expression of IGF-1, which likely is mainly responsible for the OR2AT4-mediated organ remodelling effect, constitutes an important discovery of effects that are everything but “quite modest”.

We are not sure whether the reviewer is aware of the fact that no cell culture assay, e.g. of isolated HF cell populations *in vitro*, exists that can generate meaningful hair *growth* results which come even close to the data quality and instructiveness that is generated by human HF organ culture. Instead, the hair research field has long become wary of the limited instructiveness and predictive value of such cell culture-based data, which tend to be over- or misinterpreted and are prone to generate misleading results, since hair growth reflects the net result of multiple different, interacting cell populations, which quickly lose their original *in situ* properties once ripped out of their tissue context. Thus, it is inconceivable that any *in vitro* assay could possibly “confirm” the complex organ responses we have elucidated here *ex vivo*.

Nevertheless, we have carefully considered whether there are any cell culture-based assays that might “independently confirm” key results we already have documented. The only meaningful experiment would be to isolate and culture primary outer root sheath keratinocytes (ideally from the ORS region that most strongly expresses OR2AT4) and to check whether OR2AT4 stimulation or knock-down impacts on IGF-1 production by these cells *in vitro* – provided we have first shown that these cells still express OR2AT4 after having been removed from of their physiological habitat and speculating that they still behave in a similar manner as native ORS keratinocytes, despite the fact that they have lost their crucial signaling interactions with the HF mesenchyme.

However, even if we show all this, what else would we really have proven that we don’t know already from the existing, physiologically much more relevant *in situ* data presented in our study? Moreover, what really matters in the current context is whether and how IGF-1 secreted by ORS keratinocytes impact on the apoptosis of human hair matrix keratinocytes. This question would remain unanswered since the latter cell population cannot yet be cultured. Therefore, while we are willing to run such cell culture experiments, if specifically asked to do so by the editors, we question whether they would meaningfully complement the available *in situ* data from organ-cultured HFs.

Another alternative assay method that is definitely worth considering here is the xenotransplantation of human scalp skin/hair follicles (HFs) onto immunocompromised mice (see Oh et al., 2016, Gilhar et al., 2016). However, here we are limited by applicable E.U. legislation: Since OR2AT4 agonists and antagonists clearly qualify as cosmetic agents and already form the basis of a corresponding hair growth-stimulatory product on the Italian market (“Bioscalin Signal Revolution[®]”, Giuliani/Milano), animal testing is strictly prohibited for such agents (see e.g. directive (EC) 1223/2009), and the inclusion of animal experiments in the current context would challenge the use of Sandalore[®] as a cosmetic hair growth-stimulatory agent.

[redacted]

In the meantime, we have managed to convince Giuliani S.p.A./Italy to commission a professionally executed, double-blinded, prospective, long-term clinical study that uses quantitative phototrichograms and standardized global vertex photography as objective, gold standard read-out parameters as well as an optimally chosen vehicle control (i.e. natural sandalwood oil, which smells like Sandalore[®] but does *not* activate OR2AT4 on human keratinocytes [Busse et al., 2014]). However, the results of this long-term study are expected to become available only within the first half of 2019, and will therefore be reported separately in a specialized clinical dermatology journal.

The authors mention at the beginning of the manuscript that olfactory receptors are evolutionarily ancient chemosensory systems. To my knowledge, mice have an OR2AT4 homolog which may also have a conserved role in HF growth.

Thanks for this excellent comment. OR2AT4 is one of the most homologous OR between human and mice (88% conserved), with the highest known OR homology between mice and humans being about 94% (source: The Human Olfactory Data Explorer). Therefore, it is not unreasonable to speculate that the OR2AT4 receptor in mice might be activated by the same ligands and plays a similar role in murine HF biology as it does in human scalp HFs. However, the molecular receptive field of murine OR2AT4 would first have to be rigorously tested, because the exchange of even one (of the about 320) amino acids can already suffice to change the spectrum of OR ligands strongly (Wolf et al., 2017). Therefore, it remains very uncertain and as yet entirely unproven whether the murine OR2AT4 receptor recognizes the same ligands as its human counterpart.

Apart from the legislative dilemma we face in this translational research project geared towards the development of a novel, odorant-based hair growth-stimulatory cosmeceutical (=no animal experimentation allowed!), conceptually, it is by no means plausible why an animal model should be required to increase the “credibility” of an exciting novel biological phenomenon observed in a *human* (mini-)organ. While mouse models are absolutely invaluable in biomedical research, apart from the regulatory constraints listed above, this really makes little sense here: in the current study, we are clearly not interested in the role of ORs in murine HF biology or in whether the novel mechanism we have uncovered here is conserved between mice and man (these are interesting questions for follow-up work). Instead, we have managed to silence the relevant receptor in a human organ, and have demonstrated a plausible mechanism of action (IGF-1 upregulation) without the necessity to resort to any animal model. We view this as a particularly encouraging example for incisive translational research being possible *without* having to sacrifice animals, and trust that *Nature Communications* supports such animal-sparing research whenever this is possible.

The authors should at least discuss the possibility of studying these chemoreceptors *in vivo*.

Thanks for this important suggestion. We have now developed the above line of argumentation in an extended supplementary text so as to encourage colleagues in murine olfactory research to, next, probe the effects of epithelially targeted OR2AT4 knock-out in mice, given that this would undoubtedly enrich the field as a whole and may well uncover additional roles for OR2AT4 signalling in mammalian epithelial biology (new supplementary Text 4):

“Mice have an OR2AT4 homolog which might exert a conserved role in HF growth. In fact, OR2AT4 is one of the most homologous ORs between human and mice (88% conserved), with the highest known OR homology between mice and humans being about 94% (source: The Human Olfactory Data Explorer). Therefore, it is theoretically conceivable that the murine OR2AT4 receptor might be activated by the same ligands and plays a similar role in murine HF biology as it does in human scalp HFs. However, the molecular receptive field of murine OR2AT4 would first have to be rigorously tested, because the exchange of even a single one of its about 320 amino acids can already be sufficient to change the spectrum of OR ligands recognized greatly (Wolf et al., 2017). Therefore, it remains quite uncertain and as yet entirely unproven whether the murine OR2AT4 receptor recognizes the same ligands as its human counterpart. “

Also, as a synthetic fragrance Sandalore® may already be used on skin care products. One can envision a clinical study to assess the effects of such formulations to hair growth.

The reviewer is absolutely right. Indeed a hair follicle care product has already been released in the Italian market by the company that has supported the current project. [redacted] a clinical trial is currently under way, whose results are expected at some time next year (see above).

Other comments:

1) The authors need to explain the “Hair cycle score” better because despite the citation it is difficult to follow for the non-hair expert.

Thanks (done as requested).

The hair cycle score (HCS) consists of giving an arbitrary unit for each stage of the hair cycle (Anagen VI=100; early catagen=200; Mid catagen=300 and Late catagen=400). After having classified each hair follicle according to its hair cycle stage, following the objective classification criteria for organ-cultured human HFs (Langan et al., 2015; Kloepper et al., 2010), for each experimental condition, the

mean HCS is calculated. The closer the mean is to 100, the higher is the number of anagen VI HFs in a given group. The HCS provides a global read-out parameter that looks at all HFs in a given experimental groups and synthesizes them into a single number, which reflects how close the majority of HFs is to either anagen VI or catagen and permits statistical analysis that it is not possible with hair cycle staging. Therefore, hair cycle staging and the HCS are independent read-out parameters that complement each other. This has now been explained in the revised supplementary material and methods (p. 5).

2) It is not clear to me why treatment with the antagonist Phenirat® alone does not retard hair growth further compared to vehicle. The idea is that under baseline conditions hair follicles *ex vivo* will grow to some extent and that growth is in part sustained by a still active OR2AT4. Therefore, treatment with the antagonist should impair that baseline growth.

This is a very valid question that we have much pondered ourselves. Phenirat® is known to be a competitive antagonist of OR2AT4 only in the presence of Sandalore® (Busse et al, 2014; Manteniotis et al, 2016; Pavan et al, 2017). We have now clarified this in the main text (see p.1, second paragraph). Although the results derived from the pooled data of three independent experiments (3 donors) presented in the original version of the manuscript do not appear to support the above hypothesis (see Fig. 1b), as requested by the reviewer, we have further clarified whether Phenirat® alone retards hair growth. For this purpose, we have now run 4 additional, independent *ex vivo* experiments in which HFs from 4 distinct patients were treated either with Phenirat® alone or vehicle.

This showed that Phenirat® alone tendentially promotes premature catagen development *ex vivo*, even though this did not reach significance due to the high degree of interindividual variability in human HF responses to Phenirat® seen in these experiments (see new Supplementary Fig. 4a). This new data together with our old data (see supplementary Fig. 4b) suggest that, overall, Phenirat® operates only as a weak inhibitor of human hair growth *ex vivo* in absence of Sandalore®. It is conceivable that the small difference between vehicle and Phenirat® alone may be due to the fact that high levels of (unknown) endogenous OR2AT4 ligands still present in organ-cultured control HFs did not suffice to permit Phenirat® to efficiently compete with endogenous ligands at the receptor in our model; moreover, currently available literature data do not permit one to formally exclude that Phenirat® might also bind to other ORs, which might counteract the hair growth-inhibitory effects it exerts via binding to OR2AT4. These possibilities are now explicitly mentioned in the Supplementary text 2.

3) The differences seen in Fig1e appear to be non-significant

The reviewer is correct that the differences in Fig. 1e were not statistically significant. However, we have now applied the iterative Grubbs outlier test to our data; after deleting outliers, the differences between the IGF neutralizing ab and IGF neutralizing ab + Sandalore® are significant (Fig. 1e). We also show the set of data in % of hair follicles in each hair cycle stage (Langan et al., 2015; Kloepper et al., 2010) so as to improve the visualization of the differences between the groups for Fig. 1b and e. This has now been explained in the revised legend of Fig. 1. Furthermore, for additional guidance, we have complemented the revised Discussion with the following supplementary text (supplementary text 3):

The concentration of IGF-1 neutralizing antibody used in our study has been shown to neutralize 5.0 ng/ml of IGF-1 (see datasheet of the IGF-1 neutralizing antibody, Abcam, ab9572). We have purposely used this concentration of the neutralizing antibody to neutralize only a certain % of the endogenous IGF-1 present in the

hair follicle ex vivo, to be able to have at least some hair follicles remaining in anagen after the treatment (IGF-1 is required for maintaining human HFs in anagen [17]). As one can appreciate in Fig. 1e, the selected concentration of IGF-1 neutralizing antibody was sufficient to promote premature catagen induction by reducing the concentration of available endogenous IGF-1 for the binding to the receptor (see hair cycle staging graph). Instead, Sandalore®, which increases endogenous IGF-1 protein expression and prolongs anagen (Fig. 1b, d), significantly counteracts premature catagen induction by the selected concentration of IGF-1 neutralizing antibody. Therefore, this data strongly suggests that the anagen-prolonging effect of OR2AT4 activation by Sandalore® is mainly, but possibly not exclusively, due to the increased IGF-1 secretion.

4) The phosphokinase assay suggests that the effects of Sandalore® treatments are not always dose-dependent. The authors should comment on this and also consider re-evaluating some of their experiments taking the dose into consideration. It is not clear why 500µM was chosen for all the experiments.

Given the extremely scarce availability of human scalp HFs, classical dose-response studies are generally not possible, and the test agent concentrations used must be carefully selected. The choice of 500µM was based on the previously established dose-response curve of Sandalore®-induced Ca²⁺ signals on HaCaT cells and human keratinocytes (Busse et al., 2014) where the IC₅₀ was 430µM. In this previous study, we had also shown that, at 500µM, the proliferation and migration of human epidermal keratinocytes is significantly increased compared to vehicle and that the reepithelialisation of experimental human skin wounds *ex vivo* is significantly accelerated, while concentrations below 100µM failed to activate OR2AT4 (Busse et al., 2014).

Furthermore, in additional preparatory experiments, we had compared the effects of 50 and 500µM of Sandalore® on hair matrix keratinocyte proliferation/apoptosis as well as on OR2AT4 expression; again, this demonstrated maximal effects at 500µM. These data are now documented in a new supplementary figure (Supplementary Fig. 2a-b, d) and explained in the revised supplementary materials and methods (p. 1) and text (supplementary text 1).

5) The authors discuss the possibility of microbial involvement in hair physiology through OR2AT4, based on the finding dermcidin shows the strongest up-regulation following treatment with Sandalore®. Perhaps this result is due to cross-contamination from sweat glands that were not fully dissociated from the HF during the microdissections.

The authors have previously reported on the integral relationship between sweat glands and HFs so this is not a remote possibility.

Thank you for raising this excellent point. However, this possibility can be positively excluded since, in contrast to e.g. 1 mm follicular unit punch biopsies (Poblet et al., 2016), the HF microdissection technique employed for setting up the classical Philpott assay used in the current study, removes all perifollicular tissue with the sole exception of the HF's dermal sheath, which does not contain any eccrine gland elements (Langan et al., 2015). This eliminates the theoretical contamination possibility raised by the referee. This has now been clearly explained in the revised supplementary materials and methods (p.1).

Moreover, we now show novel protein evidence that dermcidin is indeed up-regulated by Sandalore® in the epithelium of human scalp hair follicles itself (new supplementary Fig. 10). This independently supports that the up-regulation of dermcidin gene expression by Sandalore® (supplementary Fig. 8a-b and Table 1) corresponds to an intrafollicular up-regulation of this antimicrobial peptide. This new data, which further argues against the above eccrine gland contamination hypothesis and underscores that dermcidin – contrary to conventional wisdom (Burian & Schitteck 2015) - is also expressed outside of the eccrine gland (Dahlhoff et al. 2016), has now been included in the revised Results (p. 3):

6) The authors perhaps could take advantage of the OR2AT4 antibody to more accurately characterize expression levels and cellular specificity during the different stages of the hair cycle by western blot analysis and flow cytometry. For example, in their RNAi experiments a western blot would give a more accurate view than immunofluorescence.

As requested, we have now performed additional Western blot analysis of OR2AT4 protein expression in anagen VI versus catagen hair follicles. As shown in the new supplementary Fig. 1 (Supplementary Fig. 1f-g), we have successfully established Western-blot for OR2AT4 in human scalp hair follicle, which detected a specific band around 44kDa (the predicted molecular weight of OR2AT4 is 36 kDa). This difference may be explained by a post-translational modification (an acetylation site has been identified on the lysine at position 303 [source: phosphoSitePlus®]) that would increase the molecular weight (Guan K.L. et al, 2010) thus accounting for the observed difference in MW of the specific bands for OR2AT4. This is explained in the legend of our new supplementary Fig. 1 f-g.

Moreover, quantitative analysis of the protein band recognized by our OR2AT4 antibody (protein expression levels were normalized to corresponding actin levels) upon Western blotting confirmed our previous qRT-PCR and quantitative immunohistomorphometry-based observations (Supplementary Fig. 1d-e) that OR2AT4 protein expression is decreased in human catagen scalp hair follicles. These novel data are briefly stated in the revised Results (p. 1) and explained in more detail in the new supplementary Fig. 1.

Reviewer #2

The overall findings of this paper are quite interesting. There is a growing awareness that olfactory receptors can be found in numerous other tissues (the authors are not comprehensive in their citations of this) and that they have in many cases unknown functions and unknown ligands. The Sandalore® receptor is a case in point. This receptor is not activated for example by natural sandalwood but only by the synthetic Sandalore®. Its endogenous ligand in HFs is completely unknown.

As requested, we have now increased the number of supplementary citations which underscore that olfactory receptors can be found in numerous other tissues (Weber et al., 2017; Gelis et al., 2017; Pluznick et al., 2013).

The authors use a variety of controls to show that the effects are produced by the activation of the Sandalore® OR, including antagonists, knockdown and antibody blockers. In general, it appears that treatment with Sandalore® inhibits apoptosis in HF cells, although it is not clear which of several pathways are involved – or perhaps all of them.

Here, we respectfully disagree with the reviewer. While our data indeed suggest that several additional pathways may well be involved, our IGF-1 neutralizing ab experiment demonstrates that IGF-1 secretion is a *necessary* mechanism for OR2AT4 stimulation leading to prolonged anagen *ex vivo* (Fig. 1e). Moreover, specific OR2AT4 silencing in the presence of excess agonist down-regulated intrafollicular IGF-1 mRNA and protein expression. Thus, while our phosphokinase activity and microarray analysis data (Supplementary Fig. 8, 9, and 11 and Table 1), point towards the involvement of additional pathways besides IGF-1 (Supplementary Fig. 12), we have positively identified at least one distinct, IGF-1-dependent pathway that is *required* for Sandalore® to promote human hair growth *ex vivo*. This has now been clarified in our revised Discussion (p. 3-4).

There is some concern that the effects of Sandalore® require exposure to relatively high concentrations for 6 days. Under those circumstances can the authors confidently rule out non-

specific effects, possibly even membrane permeability to this volatile compound that may be at work?

Thanks for raising this important point. In our previous study (Busse et al., 2014), we had carefully addressed the specificity and receptor dependence of Sandalore® effects at the relatively high concentration tested in human HaCaT cells and primary human epidermal keratinocytes. For example, we had demonstrated that after 5 days of Sandalore® stimulation in the presence of specific antagonist of OR2AT4 or of relevant signalling pathway blockers of adenylyl cyclase (MDL-12.330A or SQ-22536) and CNG channel (L-cis Diltiazem), the Sandalore® effect in HaCaT cells or primary human epidermal keratinocytes was completely blocked, without any effects on cell morphology or any other cell biology parameter evaluated. In addition, the co-application of the specific OR2AT4 antagonist Phenirat® with Sandalore® (1:1, 500 mM) completely blocked Sandalore®-induced calcium signals in HaCaT cells indicating that the effects of Sandalore® are based on the activation of OR2AT4 in keratinocytes.

While these cell culture data demonstrated OR2AT4-dependency of the Sandalore® effects on human keratinocytes *in vitro* (for details, see Busse et al., 2014), we agree that even the Phenirat® + Sandalore® co-stimulation experiment (which showed that Sandalore® effects on the HF can be partially antagonized by Phenirat® only in the presence of Sandalore®) does not “confidently rule out” that OR2AT4-INDEPENDENT effects, possibly including even changes in membrane permeability to this volatile compound, might also have been at work, given the relatively high concentration of Sandalore® and long duration of HF organ culture.

That is exactly why we went one key step further and silenced OR2AT4 in organ-cultured HFs (incidentally, to the best of our knowledge, this is the first time that any olfactory receptor has been successfully knocked-down in any human [mini-] organ). This clearly showed that – notably *in the presence of high-dose Sandalore®!* – OR2AT4 knock-down exerted the opposite effects of agonist treatment alone and profoundly inhibited both, hair growth (i.e. OR2AT4 siRNA shortened anagen and prematurely induced catagen and up-regulated hair matrix keratinocyte apoptosis) and intrafollicular IGF-1 production.

However, to alert readers to the important caveat raised by the expert reviewer, we have complemented the revised Discussion with the following supplementary text (supplementary text 5):

“Previously, we had addressed the specificity and receptor dependence of Sandalore® effects at the relatively high concentration tested here in human HaCaT cells and primary human epidermal keratinocytes (Busse et al 2014). This had demonstrated that after 5 days of Sandalore® stimulation in the presence of specific antagonist of OR2AT4 or of relevant signalling pathway blockers (adenylyl cyclase inhibition by MDL-12.330A or SQ-22536; CNG channel inhibition by L-cis Diltiazem), the Sandalore® effect in HaCaT cells or primary human epidermal keratinocytes was completely blocked, without any effects on cell morphology or any other cell biology parameter evaluated. In addition, co-application of the specific OR2AT4 antagonist Phenirat® with Sandalore® (1:1, 500 mM) completely blocked Sandalore®-induced calcium signals in HaCaT cells (Busse et al. 2014), indicating that the effects of Sandalore® are based on the activation of OR2AT4 in keratinocytes. Most importantly, silencing OR2AT4 in organ-cultured HFs clearly showed that – notably in the presence of high-dose Sandalore® – OR2AT4 knock-down exerted the opposite effects of agonist treatment alone and profoundly inhibited both, hair growth (i.e. OR2AT4 siRNA shortened anagen and prematurely induced catagen and up-regulated hair matrix keratinocyte apoptosis) and intrafollicular IGF-1 production. “

The possibility of a hair loss treatment is enticing but the data on p53 pathway inhibition and the blocking of apoptosis suggests a potential danger of carcinogenic effects.

While we appreciate the reviewer's concern regarding potential carcinogenic effects, which must always be rigorously contemplated, especially with a novel cosmeceutical treatment strategy, a number of considerations make this concern highly implausible:

1. The decrease of apoptosis alone is insufficient to promote carcinogenesis, a complex multi-step process that necessarily requires also major genetic alterations.
2. In fact, *in vivo*, human scalp HFs stay in their growth phase (anagen VI) for many years via continuously suppressing apoptosis in the hair matrix, yet essentially never develop malignant tumors that arise from the hair matrix; once hair matrix apoptosis is no longer suppressed sufficiently, anagen HFs invariably regress and enter into catagen (Paus & Cotsarelis, 1999, Schneider et al., 2009).
3. Moreover, we are unaware of any agent that does not induce genetic instability and or is carcinogenic *per se* that has been shown in the hair research literature to inhibit apoptosis in the human hair matrix *ex vivo* or *in vivo* (e.g. minoxidil, thyroxine, finasteride, caffeine, IGF-1) to ever induce the development of malignant hair matrix tumors.
4. In previous toxicological screens (Burdock and Carabin, 2008), such as genotoxicity and cytotoxicity testing *in vitro*, and carcinogenicity test *in vivo*, Sandalore® did not generate results that support any carcinogenicity concerns; therefore, Sandalore® is not listed as carcinogenic substance by the OSHA (Occupational Safety and Health Administration (29 CFR 1910.1001-1050).

Thus, taken together, potential carcinogenicity is not a plausible concern in the current case. This is now briefly explained in an additional supplementary note (supplementary text 1).

This paper is a useful contribution to the growing literature on olfactory receptor expression in non-nasal tissue. My only objection in this regard is that there seems to me to be some loose language in this paper that confuses smelling or olfactory processes with chemical detection, which is a general cellular function. For example in the abstract, and elsewhere, the authors state that human HFs can "smell"... putting the word smell in quotes does nothing to clear up the confusion.

We agree that our choice of terminology is not always as exacting as would be expected from a professional olfactory research perspective. However, our study attempts to conceptually bridge two previously completely separate fields (olfactory and hair research) and to make key concepts in each of these two very distinct life science camps mutually understandable to a broad readership.

For example, the claim that "HFs can 'smell' in the abstract is meant to attract the attention of non-specialist readers, yet without conveying a misleading concept, if one liberally interprets "smell" to encompass OR-specific chemosensation. Throughout the manuscript, we consistently claim that human HFs engage in OR-dependent chemosensation (i.e. not just in any kind of generic chemosensation, but a very specific, odorant- and OR-dependent one), and try hard to avoid confusing smelling or olfactory processes with general chemical detection. Of course, if we underperform in this respect anywhere in the manuscript, we are happy to accept specific guidance from this referee and the editors as to where exactly we should modify our choice of terminology.

Cited references

Burdock GA and Carabin IG. (2008) Safety assessment of sandalwood oil (*Santalum album* L.). *Food Chem Toxicol.* 46(2):421-32.

Burian M and Schitteck B (2015) The secrets of dermcidin action. *Int J Med Microbiol.* 305(2):283-6.

- Busse D, Kudella P, Grüning NM, *et al.* (2014) A synthetic sandalwood odorant induces wound-healing processes in human keratinocytes via the olfactory receptor OR2AT4. *J Invest Dermatol.* 134(11):2823-32.
- Dahlhoff M, Zouboulis CC, Schneider MR. (2016) Expression of dermcidin in sebocytes supports a role for sebum in the constitutive innate defense of human skin. *J Dermatol Sci.* 81(2):124-6.
- Gelis L, Jovancevic N, Bechara FG, *et al.* (2017) Functional expression of olfactory receptors in human primary melanoma and melanoma metastasis. *Exp Dermatol.* 26(7):569-576.
- Gilhar A, Schrum AG, Etzioni A, *et al.* (2016) Alopecia areata: Animal models illuminate autoimmune pathogenesis and novel immunotherapeutic strategies. *Autoimmun Rev.* 15(7):726-35.
- Guan KL, Yu W, Lin Y, *et al.*, (2010) Generation of acetylysine antibodies and affinity enrichment of acetylated peptides. *Nat Protoc.* 5(9):1583-95.
- Kloepper JE, Sugawara K, Al-Nuaimi Y, *et al.* (2010) Methods in hair research: how to objectively distinguish between anagen and catagen in human hair follicle organ culture. *Exp Dermatol.*19(3):305-12.
- Langan EA, Philpott MP, Kloepper JE, *et al.* (2015) Human hair follicle organ culture: theory, application and perspectives. *Exp Dermatol.* 24(12):903-11.
- Manteniots S, Wojcik S, Brauhoff P, *et al.* (2016) Functional characterization of the ectopically expressed olfactory receptor 2AT4 in human myelogenous leukemia. *Cell Death Discovery.* 2:15070.
- Oh JW, Kloepper J, Langan EA, *et al.* (2016) A guide to studying human hair follicle cycling in vivo. *J Invest Dermatol.* 136(1):34-44.
- Paus R and Cotsarelis G. (1999) The biology of hair follicles. *N Engl J Med.* 341(7):491-7.
- Pavan B, Dalpiaz A. (2017) Odorants could elicit repair processes in melanized neuronal and skin cells. *Neural Regen Res.* 12(9):1401-1404.
- Philpott MP, Green MR, Kealey T. (1990) Human hair growth in vitro. *J Cell Sci.* 97 (Pt 3):463-71.
- Pluznick JL, Protzko RJ, Gevorgyan H, *et al.* (2013) Olfactory receptor responding to gut microbiota-derived signals plays a role in renin secretion and blood pressure regulation. *Proc Natl Acad Sci U S A,* 110:4410–5.
- Poblet E, Jiménez-Acosta F, Hardman JA, *et al.* (2016) Is the eccrine gland an integral, functionally important component of the human scalp pilosebaceous unit? *Exp Dermatol.* 25(2):149-50.
- Schneider MR, Schmidt-Ullrich R, Paus R. (2009) The hair follicle as a dynamic miniorgan. *Curr Biol.* 19(3):R132-42.
- Weber L, Al-Refae K, Ebbert J, *et al.* (2017) Activation of odorant receptor in colorectal cancer cells leads to inhibition of cell proliferation and apoptosis. *PLoS One.* 12(3):e0172491.
- Wolf S, Jovancevic N, Gelis L, *et al.* (2017) Dynamical binding modes determine agonistic and antagonistic ligand effects in the prostate-specific G-protein coupled receptor (PSGR). *Sci Rep.* 7(1):16007.